# Mobile Phone Addiction and Risk-Taking Behavior among Chinese Adolescents: A Moderated Mediation Model

**DOI:** 10.3390/ijerph17155472

**Published:** 2020-07-29

**Authors:** Kai Dou, Lin-Xin Wang, Jian-Bin Li, Guo-Dong Wang, Yan-Yu Li, Yi-Ting Huang

**Affiliations:** 1Department of Psychology and Research Center of Adolescent Psychology and Behavior, School of Education, Guangzhou University, Guangzhou 510006, China; psydk@gzhu.edu.cn (K.D.); wlx_psy@e.gzhu.edu.cn (L.-X.W.); 1708400055@e.gzhu.edu.cn (G.-D.W.); 2111808175@e.gzhu.edu.cn (Y.-Y.L.); 1666100038@e.gzhu.edu.cn (Y.-T.H.); 2Department of Early Childhood Education, The Education University of Hong Kong, Hong Kong, China

**Keywords:** mobile phone addiction, risk-taking behavior, self-control, sex, adolescents

## Abstract

*Objectives*: The mobile phone (MP) is an indispensable digital device in adolescents’ daily lives in the contemporary era, but being addicted to MP can lead to more risk-taking behavior. However, little is known about the mediating and moderating mechanisms underlying this relation. To address the gaps in the literature, the present study examined the idea that MP addiction is associated with reduced self-control, which further associates with increased risk-taking behavior. In addition, this study also investigated the moderation effect of adolescent sex in the association between MP addiction and self-control. *Methods*: A three-wave longitudinal study, each wave spanning six months apart, was conducted in a sample of Chinese adolescents (final *N* = 333, 57.4% girls). *Results*: Results of the moderated mediation model suggest that after controlling for demographic variables and baseline levels of self-control and risk-taking behavior, MP addiction at T1 positively predicted increased risk-taking behavior at T3 through reduced self-control at T2 for girls but not for boys. *Conclusions*: Theoretically, these findings contribute to the understanding about the working processes in the association between MP addiction and risk-taking behavior in adolescents. Practically, the results implied that boosting self-control appeared as a promising way to reduce girls’ risk-taking behavior, particularly for those who are addicted to MPs.

## 1. Introduction

Mobile phones (MPs) are ubiquitous in the contemporary era. With the advancement of high-speed internet, MPs become more versatile and play a significant role in people’s daily lives. According to a recent report, China’s MP users reached 897 million, an increase of 4.2 million compared to 2018 [1]. Adolescents constitute a major proportion of MP users in China. Although the appropriate use of MPs (e.g., looking up useful information and maintaining positive social ties) can be beneficial to adolescents (e.g., feeling higher subjective well-being) [2,3], being addicted to a MP is associated with a wide array of undesirable outcomes in adolescents [4,5,6]. Among others, a salient undesirable consequence associated with MP addiction is risk-taking behavior [7,8,9,10]. MP addiction is a type of addiction to technology and it can be defined as the uncontrolled or excessive use of mobile phones, with an inability to control craving, feeling anxious, withdrawal, and productivity loss as symptoms [11]. Although the association between MP addiction and risk-taking behavior in adolescents has been well documented, scant research has examined how and for whom MP addiction is associated with risk-taking behavior. Examining these issues may shed light on the intervention and prevention of adolescents’ risk-taking behavior. In this study, we propose that MP addiction would be related to reduced self-control which further drives them to engage in more risk-taking behavior. In addition, we also explore whether sex would play a role as well, given the sex differences in self-control [12,13,14] and MP addiction [15,16]. We examined these issues in a sample of Chinese adolescents.

### 1.1. Mobile Phone Addiction and Risk-Taking Behavior

Risk-taking behavior refers to the actions that may potentially result in adverse consequences [17,18]. Adolescents are more prone to engage in risk-taking behavior compared to their younger peers and adults [19]. According to the dual system model, the socioemotional system of adolescents develops rapidly but the cognitive control system develops relatively more slowly [19]. The rapid development of the socioemotional system enhances adolescents’ pursuit of rewarding stimuli, while the slow development of the cognitive control system limits the adolescents’ inhibition of risk-taking behavior [20,21]. The imbalance between the development of the socioemotional system and the development of the cognitive control system results in the increase in adolescents’ risk-taking behavior.

Compared to computers, the MP is more flexible, mobile and timely. Being addicted to a MP can be associated with risk-taking behavior in adolescents for two reasons [21,22]. First, individuals tend to seek rewarding incentives, novel stimuli and excitement through engaging in risk-taking behavior, common in adolescents [19]. The use of a MP is often accompanied with rewarding incentives, novel stimuli and excitement, and thus MP addiction may exacerbate the adolescents’ tendencies of engaging in risk-taking behavior [23,24]. Previous research has indicated that adolescents who frequently use MPs engage in more reckless driving [25], smoking and alcohol abuse [26] than those who are not addicted to MPs. Second, MP addiction may restrain adolescents’ cognitive control, which further contributes to their risk-taking behavior. In accordance with this perspective, a previous study has found that those who are addicted to MPs are reluctant to spend cognitive resources and are prone to adopt intuitive thinking in daily lives [27,28]. In the absence of cognitive control and analytical thinking, individuals may have more difficulties in restraining the tendencies of seeking novelty, rewarding incentives, and excitement through risk-taking behavior [29,30,31]. Taken together, we assume that MP addiction would be related to an increase in risk-taking behavior in adolescents.

### 1.2. The Mediation Effect of Self-Control

Self-control is defined as the ability that individuals make effort to overcome impulsion and automatic reaction, and to support the pursuit of long-term goals [32,33]. As a vital psychological function, self-control is associated with a number of positive outcomes, including less risk-taking behavior [32,33,34]. The general theory of crime postulates that individuals with low self-control are inclined to be short-sighted and impulsive, which is a core cause of young people’s delinquency [35]. Research has found that adolescents with low self-control are more likely to have excessive drinking [36,37], substance abuse [38], gambling [39,40] and other risk-taking behavior. In addition, the strength model of self-control posits that self-control resource depletion in the previous stage limits the availability of self-control for the next stage, which may increase the likelihood of the occurrence of risk-taking behavior [41,42].

Self-control may play a “bridge” role between MP addiction and adolescents’ risk-taking behavior. There could be different pathways from mobile phone addiction to low self-control. On the one hand, MP addiction may reduce cognitive control, distract attention, and make the cognitive control system “lazied” in adolescents, and thus they prefer intuitive cognitive processing [27,28,43]. For instance, in a sample of 1721 adolescents, Hong et al. (2020) found that MP addiction leads to cognitive failures. On the other hand, MPs may provide immediate stimulation and feedback that may activate the socioemotional system, rendering adolescents vulnerable to instant gratification and short-term rewards. Individual differences in low self-control and the temporary depletion of self-control resources due to MP addiction can render adolescents’ cognitive resources insufficient to override the tendencies of seeking novelty and excitement, which may be further associated with more risk-taking behavior. On these bases, we assume that MP addiction can indirectly affect adolescents’ risk-taking via reduced self-control.

### 1.3. The Moderation Effect of Sex

Previous research has revealed sex differences in the pattern of mobile phone use [15,16]. For instance, girls are more prone to use MPs for social networking, entertainment and shopping [44,45], while boys are more likely to use MPs for work and games [15]. In this sense, compared to boys, girls are considered to be more emotionally involved when using MPs, experience more emotional swifts, and have higher social motivation [15]. In addition, boys are prone to engage in more risk-taking behavior than girls during adolescence [46], because boys have higher sensation-seeking tendencies and lower impulsive control [47,48]. As a relatively malleable personal characteristic, self-control can be affected by environment. Compared to boys, girls are more likely to be susceptible to external factors (e.g., MP addiction) [13]. In line with this, MP addiction could suppress girls’ self-control rather than boys. Therefore, we assume that sex would moderate the effect of MP addiction on adolescents’ self-control.

### 1.4. The Present Study

Taken together, this three-wave longitudinal study, with each wave spanning six months apart, investigates the association between MP addiction and risk-taking behavior as well as the underlying mechanisms in a sample of Chinese adolescents. Specifically, we would examine the idea that MP addiction would be associated with increased risk-taking behavior through reduced self-control. Moreover, we would examine the moderating role of sex (see in Figure 1). In sum, we hypothesized that: (1) MP addiction would be positively related with adolescent risk-taking behavior; (2) self-control would mediate the relation between MP addiction and risk-taking behavior; (3) sex would moderate the effect of MP addiction on adolescents’ self-control, with the negative effect of MP addiction on self-control being stronger for girls than boys; and (4) sex would moderate the mediation effect of self-control, with the mediation effect of self-control being more pronounced for girls than boys. Combining all these hypotheses results in a moderated mediation model (Figure 1).

## 2. Method

### 2.1. Participants and Procedures

The data were collected from a public middle school in a large city in southern China. All the procedures involving human participants were reviewed and approved by the research ethics committee in the School of Education at Guangzhou University (Protocol Number: GZHU2019018). Written consent forms from the parents and oral assent from the adolescents were obtained before data collection across the waves. At each wave, two trained research assistants hosted the survey and the participants completed the questionnaires during regular class hours in the classroom. All the participants received a small gift worthy of 15 RMB (approximately 2.5 US Dollars) after completing the questionnaires each time.

A total of 412 parents provided consent for their children’s participation. Finally, 399 10th graders (M = 15.37, SD = 0.52, 52.1% girls) participated in the first wave of data collection (Time 1, T1). Of the 399 adolescents, 353 (attrition rate = 11.53%) and 386 (attrition rate = 3.26%) participated in the assessments at Time 2 (T2) and Time 3 (T3), respectively. The time interval of the data collection between each wave was six months. Detailed demographic characteristics of the T1 sample are presented in Table 1.

### 2.2. Measures

#### 2.2.1. Mobile Phone Addiction at T1

We used the Mobile Phone Addiction Index Scale (MPAI) [49] to measure the participants’ frequency of using MPs at T1. This scale consists of 17 items rated on a five-point scale (from 1 = never done to 5 = almost always). A mean score can be calculated by averaging all the items, with a higher score indicating more the frequent use of MP. Sample items are “Your friends and family complained about your use of the mobile phone” and “You feel lost without your mobile phone”. In the current study, the Cronbach’s alpha of this scale was 0.97.

#### 2.2.2. Self-Control at T1 and T2

We used the Chinese version of Tangney et al.’s (2004) Brief Self-Control Scale (BSCS) [50,51] to assess the participants’ self-control ability at T1 and T2. This scale consists of 13 items rated on a five-point scale (from 1 = not like me at all to 5 = like me very much). A higher mean score indicates a better self-control ability. The sample items are “I am good at resisting temptation” and “Sometimes I can’t stop myself from doing something, even if I know it is wrong”. In this research, the Cronbach’s alpha of this scale at T1 and T2 was 0.79 and 0.80, respectively.

#### 2.2.3. Risk-Taking Behavior at T1, T2, and T3

Risk-taking behavior was assessed by the 15-item Adolescent Risk-Taking Questionnaire (ARQ) [17] through T1 to T3. Adolescents reported the frequency of performing various risk-taking behavior (e.g., unprotected sex, driving/cycling after drinking) on a five-point Likert scale (from 0 = never done to 4 = done very often). A mean score was calculated by averaging all the items. This measure has been translated into Chinese and demonstrated to be valid and reliable in Chinese samples [18]. In this study, Cronbach’s alpha of this scale at T1, T2 and T3 was 0.96, 0.81 and 0.76, respectively.

#### 2.2.4. Covariates at T1

The child’s age, only child at home or not (0 = Yes, 1 = No), parents’ employment status (1 = freelance, 2 = par-time job, 3 = full-time job) and educational levels (1 = junior middle school and below, 2 = high school degree, 3 = college degree, 4 = bachelor’s degree, 5 = master’s degree or doctoral degree) were included as covariates since prior studies have found significant associations between these demographic variables with risk-taking behavior [52,53].

### 2.3. Data Analyses

Initially, descriptive statistics and bivariate correlations were performed using SPSS 22.0 ((IBM, Armonk, NY, USA) to examine the centrality and association among the variables of interest. Second, structural equation modeling (SEM) was performed using Mplus 7.0 (Muthén & Muthén, Los Angeles, CA, USA) to test the hypothesized moderated mediation model. The missing data were handled with the full information maximum likelihood estimation (FIML) [54]. In this model, T1 MP addiction was the independent variable; T2 self-control was the mediator; T3 risk-taking behavior was the outcome; and sex was the moderator. In this model, we also controlled for the baseline levels of self-control at T1 and risk-taking behavior at T1 and T2, as well as the effect of covariates on the outcome (i.e., T3 risk-taking behavior). Given that the bootstrapping technique has several advantages over the traditional approaches in examining mediation models such as higher statistical power [55], we used bootstrapping (*N* = 5000) and its 95% confidence intervals to judge the significance of the mediation. As long as the 95% confidence interval excludes 0, significant mediation effect is tenable. The following indices were used to evaluate the overall model fit [56]: a nonsignificant chi-square statistics (*χ*^2^), the comparative fit index (CFI), the root mean square error of approximation (RMSEA) [57] with its 90% confidence interval (CI), and the standardized root mean square residual (SRMR). However, given that the sample size of the current study is large and the *χ*^2^ statistic is sensitive to sample size, a significant *χ*^2^ statistic was expected.

## 3. Results

### 3.1. Descriptive Statistics and Bivariate Correlation

Means, standard deviations, and bivariate associations are shown in Table 2. As can be seen in the table, MP addiction at T1 was negatively related to T1 and T2 self-control (*r* = −0.43 and −0.35, *ps* < 0.001), but positively associated with T1/T2/T3 risk-taking behavior (*r* = 0.22–0.37, *ps* < 0.001). Adolescent self-control and risk-taking behavior were also negatively correlated, within and across time points (*r* = −0.22–−0.30, *ps* < 0.001). According to Cohen’s (1992) standard [58], the effect sizes of these correlation coefficients were small-to-medium.

### 3.2. Examination of the Hypothesized Moderated Mediation Model

The hypothesized moderated mediation model was examined. The fit indices were *χ*^2^ = 36.91, *df* = 22, *p* < 0.05, RMSEA = 0.04 (90% CI = [0.015, 0.064]), CFI = 0.959, and SRMR = 0.039, indicating that the model was a good fit. This model accounted for the 20.5% variance of “risk-taking behavior”, and the corresponding effect size was medium to large (*f* = 0.26) [58].

As shown in Figure 2 and Table 3, T1 MP addiction was not directly related to T3 risk-taking behavior at the statistically significant level. Nevertheless, the T1 MP addiction was significantly related to T2 self-control (*B* = −0.08, *SE* = 0.03, *p* = 0.009). Moreover, T2 self-control was significantly related to T3 risk-taking behavior (*B* = −0.11, *SE* = 0.04, *p* = 0.003). More importantly, T2 self-control significantly linked the association between T1 MP addiction and T3 risk-taking behavior (*B* = 0.01, 95% CI = [0.002, 0.021]), but not the effect of baseline levels of self-control, risk-taking behavior, and covariates.

Regarding the moderation effect of sex, there is a significant interaction effect between MP addiction and adolescent sex on T2 self-control (*B* = −0.13, *SE* = 0.06, *p* = 0.02). As shown in Figure 3 and Table 4, the association between T1 MP addiction and T2 self-control was significant only for girls (*B* = −0.14, *SE* = 0.04, *p* < 0.001), but not for boys (*B* = −0.03, *SE* = 0.04, *p* = 0.56). Moreover, we found that the mediation of T2 self-control was significant for girls (*B* = 0.018, *SE* = 0.007, 95% CI = [0.006, 0.036]) but not for boys (*B* = 0.004, *SE* = 0.005, 95% CI = [−0.006, 0.016]).

## 4. Discussion

MP has become an inseparable part of adolescents’ life, but MP addiction can be related to various undesirable outcomes such as high-stake risk-taking behavior. To examine how and for whom MP addiction is related to risk-taking behaviors in adolescents, this three-wave longitudinal study examines the mediation role of self-control and the moderation role of sex in a sample of Chinese high school students. The results reveal that adolescents’ MP addiction is related to increased risk-taking behavior via self-control, but this mediation model appears as only significant for girls but not for boys.

### 4.1. MP Addiction and Adolescents’ Risk-Taking Behavior

Prior studies have found that MP addiction is related to risk-driving behavior [25,59]. The current study adds to this line of literature. Supporting our first hypothesis, this study reveals that MP addiction is related to other forms of risk-taking behavior in addition to risk driving. More importantly, using a three-wave longitudinal study and controlling for the baseline levels of risk-taking behavior, our results indicate that MP addiction increases risk-taking behavior over time, although the effect is indirect rather than direct. MPs can be used for multiple content categories, such as gathering information, playing games, and maintaining social networking. Different content categories can be related to different consequences. For example, exposure to risk-taking photos that are posted on the internet may increase adolescents’ acceptance and propensity of engaging in risk-taking behavior [60,61]. However, the current study does not examine which content category is most related to risk-taking behavior. This could be a promising avenue for future research to explore.

### 4.2. The Mediation Effect of Self-Control

Confirming the second hypothesis, the results of the mediation model show that self-control mediates the effect of MP addiction on risk-taking behavior. The first part of the mediation process (i.e., mobile phone addiction → self-control) supports the flow theory. The flow theory suggests that immediate gratification and intrinsic rewards induced by mobile phones render individuals to lose themselves in electronic devices [62]. The perception of the presence of MPs is a temptation, even when people are not using it, because it distracts individuals’ attention and increases the difficulty for individuals to focus on a task [9,63]. MP addiction may lead adolescents to develop a bad habit of checking their mobile phone frequently in daily life, which may undermine adolescents’ self-control ability [64] and render them vulnerable to immediate rewards [8]. The second part of the mediation process (i.e., self-control → risk-taking behavior) is consistent with previous findings [42,65]. Self-control is a crucial psychological function associated with numerous life outcomes [34,35]. The mediation model suggests that MP addiction may increase risk-taking behavior by limiting one’s self-control ability. We call this a “restraining path”, such that MP addiction increases risk-taking behavior by restraining protective factors such as self-control. Although the current study does not provide a direct examination, it is worthwhile to note that according to the dual system model, there may be also a “promotive path” that may explain how MP addiction increases risk-taking behavior. For instance, MP addiction may increase adolescents’ sensitivity to incentives and rewarding stimuli [10,21], and thus adolescents may meet their desires by engaging in more risk-taking behavior. As this study does not examine this assumption directly, future research may examine other mediators (e.g., increased sensitivity) in the association between MP addiction and risk-taking behavior.

### 4.3. The Moderation of Sex

Confirming the third and the fourth hypotheses, our results reveal that the mediation of self-control in the association between MP addiction and risk-taking behavior is only significant for girls. This could be because girls’ self-control is more sensitive to environmental stimuli (e.g., cell phone) and more malleable [13] compared to boys, and thus MP addiction imposes more negative effect on self-control in girls than boys. As discussed above, the current study provides support to the “restraining path” and finds that this working mechanism only works among girls. Specifically, given that girls’ self-control is more malleable and matures earlier [13], girls’ self-control can sever as a protective factor of risk-taking behavior. However, MP addiction, as a risk factor, may restrain girls’ self-control, thus increasing the likelihood of risk-taking behaviors. In contrast, boys generally have higher sensation-seeking and impulsivity than girls during adolescence, which indicates that boys’ self-control cannot play its role well in prohibiting risk-taking behaviors [46]. In this case, the “restraining path” of self-control plays a little role in the association between MP addiction and the increase in risk-taking behaviors in boys. Thus, we suspect whether the “promotive path” works equally well in both sexes. Future research may examine this line of research to further deepen the working mechanisms underlying the “MP addiction–risk-taking behavior” link.

### 4.4. Implications

This study bears two implications for the prevention and intervention of adolescents’ risk-taking behavior. On the one hand, the results show that MP addiction may increase risk-taking behavior over time. This implies that addressing MP addiction may be a crucial way to reduce the occurrence of risk-taking behavior in adolescents. On the other hand, reduced self-control significantly mediates the association between MP addiction and risk-taking behavior in girls. This suggests that using evidence-based programs (e.g., mindfulness) to boost girls’ self-control may be useful in reducing girls’ risk-taking behavior.

### 4.5. Limitations

We must acknowledge that this study has several limitations. First, only self-reported data were collected and thus the associations could be inflated because of the common method bias. To enhance the internal validity of the results, future research may use multiple measurement modalities to triangulate each variable. In addition, as discussed above, this study does not reveal which content category of MP addiction is related to risk-taking behavior. Future research may deepen this issue to achieve a fuller understanding of the relationship between MP addiction and risk-taking behavior in adolescents. Finally, family relationship has been found to be associated with adolescents’ screen behaviors and risk behaviors [11]. Future research should take family relationship into further consideration.

## 5. Conclusions

Taken together, this study reveals that MP addiction is a risk factor for risk-taking behavior via reduced self-control in adolescent girls. These findings bear important implications for the prevention and intervention of adolescents’ risk-taking behavior and to the promotion of positive youth development.

## Figures and Tables

**Figure 1 ijerph-17-05472-f001:**
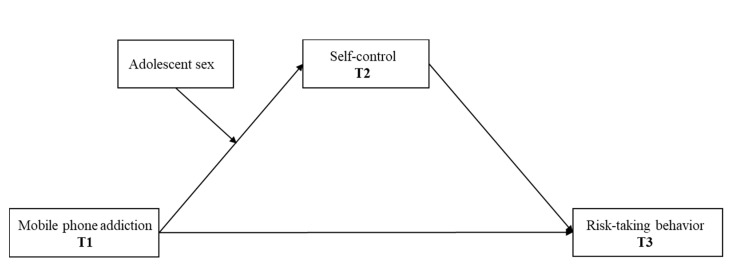
Conceptual moderated mediation model of the association between mobile phone addiction and risk-taking behavior; T1 = Time 1; T2 = Time 2; T3 = Time 3.

**Figure 2 ijerph-17-05472-f002:**
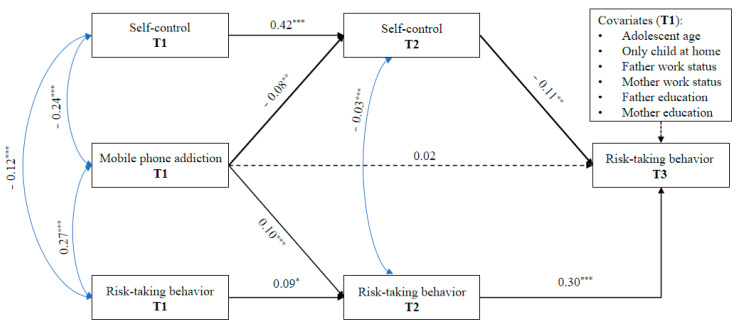
The mediation model of self-control in the association between mobile phone addiction and risk-taking behavior. Note: unstandardized estimates are presented; *χ*^2^ = 36.91, *df* = 22, *p* < 0.05, root mean square error of approximation (RMSEA) = 0.041 with 90% CI [0.015, 0.064], comparative fit index (CFI) = 0.959, standardized root mean square residual (SRMR) = 0.039; * *p* < 0.05, ** *p* < 0.01, *** *p* < 0.001. Dashed line indicates a non-significant coefficient.

**Figure 3 ijerph-17-05472-f003:**
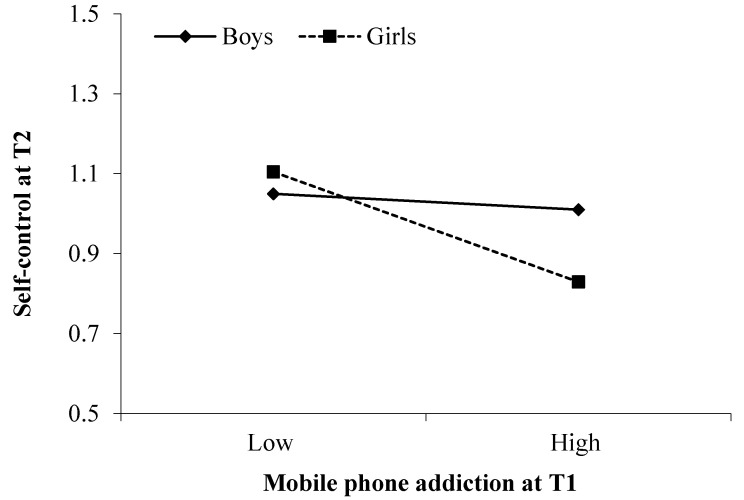
Moderation effect of adolescent sex on the link from T1 mobile phone addiction to T2 self-control.

**Table 1 ijerph-17-05472-t001:** Summary of the demographic variables at T1.

Variables	*N*	%
Adolescent sex		
Boys	191	47.9%
Girls	208	52.1%
Only child at home		
Yes	198	49.6%
No	201	50.4%
Father’s work status		
Unemployment	32	8.0%
Part-time job	45	11.3%
Full-time job	322	80.7%
Mother’s work status		
Unemployment	82	20.6%
Part-time job	53	13.3%
Full-time job	264	66.2%
Father’s highest educational level		
High school or below	234	58.6%
College or undergraduate	152	38.1%
Master or above	13	3.3%
Mother’s highest educational level		
High school or below	254	63.7%
College or undergraduate	140	35.1%
Master or above	5	1.3%
Total	399	100%

**Table 2 ijerph-17-05472-t002:** Descriptive statistics and bivariate correlations among the study variables and covariates.

Study Variables	1	2	3	4	5	6	7	8	9	10	11	12	13
1. MP addiction (T1)	——												
2. Self-control (T1)	−0.43 ***												
3. Self-control (T2)	−0.35 ***	0.51 ***											
4.Risk-taking behavior (T1)	0.37 ***	−0.28 ***	−0.22 ***										
5.Risk-taking behavior (T2)	0.32 ***	−0.25 ***	−0.30 ***	0.27 ***									
6.Risk-taking behavior (T3)	0.22 ***	−0.22 ***	−0.27 ***	0.19 ***	0.38 ***								
7. Sex (0 = boys, 1 = girls)	−0.17 ***	0.06	0.01	−0.24 ***	−0.19 ***	−0.13 **							
Covariates at T1													
8. Age	0.02	0.01	0.07	−0.03	−0.03	0.04	0.00						
9. Only child at home	−0.16 **	0.03	0.10	−0.02	−0.11 *	−0.04	0.20 ***	−0.01					
10. Father’s work status	0.02	0.07	0.01	−0.01	0.06	0.04	−0.09	0.04	−0.03				
11. Mother’s work status	0.03	−0.07	−0.06	0.05	0.06	0.01	−0.06	0.11 *	−0.09	0.23 ***			
12. Father’s education	0.13 **	−0.05	−0.11 *	0.14 **	0.01	0.02	−0.00	−0.10 *	−0.18 ***	0.07	−0.00		
13. Mother’s education	0.05	−0.05	−0.03	0.19 ***	0.08	−0.03	−0.02	−0.09	−0.17 **	0.02	0.11 *	0.56 ***	
M	2.16	3.05	3.19	1.41	0.28	0.27	0.52	15.37	1.50	2.73	2.46	2.53	2.38
SD	0.99	0.55	0.54	0.73	0.37	0.33	0.50	0.52	0.50	0.60	0.81	1.02	0.95

Note: *N* ranges from 353 to 399. MP addiction = mobile phone addiction; T1 = Time 1; T2 = Time 2; T3 = Time 1. * *p* < 0.05, ** *p* < 0.01, *** *p* < 0.001.

**Table 3 ijerph-17-05472-t003:** Regression model for the moderation effect of adolescent sex.

	Self-Control (T2, *R*^2^ = 0.29)			Risk-Taking Behavior (T2, *R*^2^ = 0.13)			Risk-Taking Behavior (T3, *R*^2^ = 0.20)		
	*B*	*SE*	*p*	*B*	*SE*	*p*	*B*	*SE*	*p*
Covariates at T1									
Age							0.03	0.03	0.41
Only child at home							−0.03	0.02	0.19
Father’s work status							0.01	0.02	0.56
Mother’s work status							0.01	0.03	0.92
Father’s education							0.01	0.04	0.80
Mother’s education							−0.01	0.02	0.77
Independent variable									
MP addiction (T1)	−0.03	0.05	0.48	**0.10**	**0.02**	**<0.001**	0.02	0.02	0.37
Mediating variable									
Self-control (T1)	**0.40**	**0.06**	**<0.001**						
Self-control (T2)							**−0.11**	**0.04**	**0.002**
Moderating variable									
Sex	0.23	0.14	0.10						
Interaction term									
MP addiction (T1) × Sex	**−0.13**	**0.06**	**0.02**						

Note: only child at home: 0 = yes, 1 = no; work status: 1 = freelance, 2 = par-time job, 3 = full-time job; education: 1 = junior middle school and below, 2 = high school degree, 3 = college degree, 4 = bachelor’s degree, 5 = master’s degree or doctoral degree; sex: 0 = boys, 1 = girls; MP addiction = mobile phone addiction; T1 = Time 1; T2 = Time 2; T3 = Time 3. Significant results are in bold.

**Table 4 ijerph-17-05472-t004:** The mediation effect of self-control by sex.

Sex	*B*	*S.E.*	95% CI
Boys	0.004	0.005	[−0.006, 0.016]
Girls	**0.018**	**0.007**	**[0.006, 0.036]**

Note: significant results are in bold.

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
