# Peer review of "Mobile Phone Addiction and Risk-Taking Behavior among Chinese Adolescents: A Moderated Mediation Model"

_ijerph, 2020, doi:10.3390/ijerph17155472_

Round 1
Reviewer 1 Report
The present study presents a very interesting and relevant topic at present.
In my opinion, the work has been very well presented and justified, however, I would like to suggest a few brief changes:
The section "The present study" seems little worked, that is to say, normally, in this section, the importance of the study is justified, as well as, the objectives are named and the hypotheses are enumerated.
It would be interesting if the authors wrote the objective in that section instead of in the introduction section.
I suggest the same for hypotheses. As they are named throughout the text, they could also be listed in the paragraph "The present study".
In the discussion, only the hypotheses H3 and H4 are mentioned (paragraph before the implications section). It would be interesting for the authors to mention all the hypotheses proposed in the study, as well as, say if they are confirmed with the results (it seems that they are), or on the contrary, no.
Good job!!!
Author Response
Response to Reviewer 1 Comments
The present study presents a very interesting and relevant topic at present.
In my opinion, the work has been very well presented and justified, however, I would like to suggest a few brief changes:
[Response] Thank you for reviewer’s time and effort devoted to our manuscript. We have made revisions in the manuscript according to the comments. All the revisions are highlighted in red in the text. Please see our response point by point below.
[Comment 1]: The section "The present study" seems little worked, that is to say, normally, in this section, the importance of the study is justified, as well as, the objectives are named and the hypotheses are enumerated
It would be interesting if the authors wrote the objective in that section instead of in the introduction section.
I suggest the same for hypotheses. As they are named throughout the text, they could also be listed in the paragraph "The present study".
[Response] Thank you for this comment. As suggested, we have added the objectives and hypotheses in the section “The present study”. These revisions are presented below.
[Taken together, this three-wave longitudinal study, with each wave spanning six months apart, investigates the association between MP addiction and risk-taking behavior as well as the underlying mechanisms in a sample of Chinese adolescents. Specifically, we would examine the idea that MP addiction would be associated with increased risk-taking behavior through reduced self-control. Moreover, we would examine the moderation role of sex (see in Figure 1). In sum, we hypothesize that: (1) MP addiction would be positively related with adolescent risk-taking behavior; (2) self-control would mediate the relation between MP addiction and risk-taking behavior; (3) sex wold moderate the effect of MP addiction on adolescents’ self-control, with the negative effect of MP addiction on self-control would be stronger for girls than boys; and (4) sex would moderate the mediation effect of self-control, with the mediation effect of self-control would be more pronounced in girls than boys. Combining all these hypotheses result in a moderated mediation model (Fig. 1).] (line 113)
[Comment 2]: In the discussion, only the hypotheses H3 and H4 are mentioned (paragraph before the implications section). It would be interesting for the authors to mention all the hypotheses proposed in the study, as well as, say if they are confirmed with the results (it seems that they are), or on the contrary, no.
[Response] Thank you for this comment. The findings support the first (line 243)and the second (line 255) hypotheses in that MP addition is associated with higher levels of risk-taking behavior and this association is mediated by insufficient self-control.
Good job!!!
[Response] Thank you for your encouragement!
Reviewer 2 Report
- An in-depth review of the writing style and use of English should be carried out throughout the text.
- The title should be changed. I recommend focusing on the concept of risk-taking behavior.
- The concept of risk-taking behavior is more complex than the authors reflect in the quote in line 48. The cited reference itself explains that the concept can be studied from different perspectives. The concept should be better defined or determine from which perspective it is understood in the present work. Later, the adolescent cognitive system is discussed, understanding a cognitive development perspective, but the focus of the work both in its theoretical approach and in the data collection methodology focuses more on the perspective of social development.
- What do authors understand by MP addiction? An objective and operational definition of MP addiction must be provided and differentiated from abusive use.
- Effect size must be provided.
- Remove reference intervals of the model fit indices from the data analysis section.
Author Response
Response to Reviewer 2 Comments
Thank you for your time and effort devoted to our manuscript. We have made revisions in the manuscript according to the comments. All the revisions are highlighted in red in the text. Please see our response point by point below.
[Comment 1]: An in-depth review of the writing style and use of English should be carried out throughout the text.
[Response] Thank you for this comment. As your suggestion, we already carefully edited the paper and eliminated the grammatical errors in the revision.
[Comment 2]: The title should be changed. I recommend focusing on the concept of risk-taking behavior.
[Response] Thank you for this comment. We have changed our title to “Mobile Phone Addiction and Risk-Taking Behavior among Chinese adolescents: A Moderated Mediation Model”.
[Comment 3]: The concept of risk-taking behavior is more complex than the authors reflect in the quote in line 48. The cited reference itself explains that the concept can be studied from different perspectives. The concept should be better defined or determine from which perspective it is understood in the present work. Later, the adolescent cognitive system is discussed, understanding a cognitive development perspective, but the focus of the work both in its theoretical approach and in the data collection methodology focuses more on the perspective of social development.
[Response] Thank you for this comment. Regarding the first question, inspired by your comment and prior research (e.g., Gullone et al., 2000; Ju et al., 2020), we have redefined the concept of risk-taking behavior by making it explicit in the revision that “Risk-taking behavior refers to the actions that may potentially result in adverse consequences [17, 18]. (line 53)
Regarding the second question, it is worth discussing. In the previous research of self-control, self-control is an umbrella construct, including delaying gratification, cognitive control, emotional control and impulse control [32,33]. There could be different pathways from mobile phone addiction to low self-control. On one hand, mobile phone may provide immediate stimulation and feedback that may activate socioemotional system, rendering adolescents to prefer instant gratification and short-term rewards. On the other hand, mobile phone addiction may reduce cognitive control, distract attention, and make cognitive control system “lazied” in adolescents. In this sense, mobile phone addiction is supposed to be negatively related to less self-control in adolescents from the perspective of socioemotional system and cognitive control system. These revisions are presented below.
[Self-control is defined as the ability that individuals make effort to overcome impulsion and automatic reaction, and to support the pursuit of long-term goals [32, 33]. As a vital psychological function, self-control is associated with a number of positive outcomes, including less risk-taking behavior [32–34]. ] (line 77)
[There could be different pathways from mobile phone addiction to low self-control. On one hand, MP addiction may reduce cognitive control, distract attention, and make cognitive control system “lazied” in adolescents, and thus they prefer intuitive cognitive processing [43–45]. For instance, in a sample of 1721 adolescents Hong et al (2020) found that MP addiction leads to cognitive failures. On the other hand, MP may provide immediate stimulation and feedback that may activate the socioemotional system, rendering adolescents to prefer instant gratification and short-term rewards. Individual differences in low self-control and temporary depletion of self-control resource due to MP addiction can render adolescents to have insufficient cognitive resources in overriding the tendencies of seeking novelty and excitement through engaging in risk-taking behavior. On these bases, we assumes that MP addiction can indirectly affect adolescents’ risk-taking via reduced self-control] (line 88)
Reference
- Gullone, E.; Moore, S.; Moss, S.; Boyd, C., The adolescent risk-taking questionnaire. J. Adolesc. Res. 2000, 15, 231–250. doi:10.1177/0743558400152003
- Ju, C.; Wu, R.; Zhang, B.; You, X.; Luo, Y., Parenting style, coping efficacy, and risk-taking behavior in Chinese young adults. J. Pac. Rim Psychol. 2020, 14, 1–9. doi:10.1017/prp.2019.24
- Moffitt, T.E.; Arseneault, L.; Belsky, D.; Dickson, N.; Hancox, R.J.; Harrington, H.; Houts, R.; Poulton, R.; Roberts, B.W.; Ross, S.; et al., A gradient of childhood self-control predicts health, wealth, and public safety. Proc. Natl. Acad. Sci. U. S. A. 2011, 108, 2693–2698. doi:10.1073/pnas.1010076108
- De Ridder, D.T.; Lensvelt-Mulders, G.; Finkenauer, C.; Stok, F.M.; Baumeister, R.F., Taking stock of self-control: a meta-analysis of how trait self-control relates to a wide range of behaviors. Pers. Soc. Psychol. Rev. 2012, 16, 76–99. doi:10.1177/1088868311418749
- Vazsonyi, A.T.; Mikuška, J.; Kelley, E.L., It's time: A meta-analysis on the self-control-deviance link. J. Crim. Justice 2017, 48, 48–63. doi:10.1016/j.jcrimjus.2016.10.001
- Barr, N.; Pennycook, G.; Stolz, J.A.; Fugelsang, J.A., The brain in your pocket: Evidence that smartphones are used to supplant thinking. Comput. Hum. Behav. 2015, 48, 473–480. doi:10.1016/j.chb.2015.02.029
- Hadlington, L.J., Cognitive failures in daily life: Exploring the link with Internet addiction and problematic mobile phone use. Comput. Hum. Behav. 2015, 51, 75–81. doi:10.1016/j.chb.2015.04.036
- Hong, W.; Liu, R.D.; Ding, Y.; Sheng, X.; Zhen, R., Mobile phone addiction and cognitive failures in daily life: The mediating roles of sleep duration and quality and the moderating role of trait self-regulation. Addict. Behav. 2020, 107, 106383. doi:10.1016/j.addbeh.2020.106383
[Comment 4]: What do authors understand by MP addiction? An objective and operational definition of MP addiction must be provided and differentiated from abusive use.
[Response] Thank you for this comment. Yu and Sussman’s (2020) meta-analytic research has indicated that mobile phone addiction is an emerging type of addiction regarding mobile phone specifically through mobile phone device. According to prior work, we defined mobile phone addiction as a type of technological addiction, including uncontrolled or excessive use of mobile phones. Mobile phone addiction can be reflected as several addictive symptoms, with inability to control craving, feeling anxious, withdrawal and productivity loss. (line 41) Abusive use belongs to the Inability to Control Craving dimension of mobile phone addiction and it is part of the components of mobile addiction.
Reference
Yu, S.; Sussman, S., Does Smartphone Addiction Fall on a Continuum of Addictive Behaviors? Int. J. Environ. Res. Public Health 2020, 17. doi:10.3390/ijerph17020422
[Comment 5]: Effect size must be provided.
[Response] Thank you for this comment. As suggested, we have provided the effect size in the section of result. These revisions are presented below.
[Adolescent self-control and risk-taking behavior were also negatively correlated (r = -0.22~ -0.30, ps < 0.001). According to Cohen’s (1992) standard [61], the effect sizes of these correlation coefficients were small-to-medium] (line 194)
[This model accounted for 20.5% variance of "risk-taking behavior", and the corresponding effect size is medium-to-large (f 2=0.26).] (line 201)
Reference
- Cohen, J., A power primer. Bull. 1992, 112, 155-159. doi:10.1037/0033-2909.112.1.155.
[Comment 6]: Remove reference intervals of the model fit indices from the data analysis section.
[Response] Thank you for the keen observation. As suggested, we have removed reference intervals of the model fit in the section of data analysis (line 184).
Round 2
Reviewer 2 Report
The authors correctly made the corrections suggested in the previous review.